# The Role of Natural Products and Their Multitargeted Approach to Treat Solid Cancer

**DOI:** 10.3390/cells11142209

**Published:** 2022-07-15

**Authors:** Naoshad Muhammad, Darksha Usmani, Mohammad Tarique, Huma Naz, Mohammad Ashraf, Ramesh Raliya, Shams Tabrez, Torki A. Zughaibi, Ahdab Alsaieedi, Israa J. Hakeem, Mohd Suhail

**Affiliations:** 1Department of Radiation Oncology, School of Medicine, Washington University, Saint Louis, MO 63130, USA; nmuhammad@wustl.edu; 2G-Bioscience, 9800 Page Ave., St. Louis, MO 63132, USA; darksha97@gmail.com; 3Department of Child Health, University of Missouri, Columbia, MO 65211, USA; tariqueunmatched@gmail.com; 4Department of Internal Medicine, University of Missouri, Columbia, MO 65211, USA; huma.7biotech@gmail.com; 5Department of Chemistry, Bundelkhand University Jhansi, Jhansi 284128, Uttar Pradesh, India; ashrafm035@gmail.com; 6IFFCO Nano Biotechnology Research Center, Kalol 382423, Gujarat, India; rameshraliya@iffco.in; 7King Fahd Medical Research Center, King Abdulaziz University, Jeddah 21589, Saudi Arabia; shamstabrez1@gmail.com (S.T.); taalzughaibi@kau.edu.sa (T.A.Z.); 8Department of Medical Laboratory Sciences, Faculty of Applied Medical Sciences, King Abdulaziz University, Jeddah 21589, Saudi Arabia; aalsaieedi@kau.edu.sa; 9Vaccines and Immunotherapy Unit, King Fahd Medical Research Center, King Abdulaziz University, Jeddah 21589, Saudi Arabia; 10Department of Biochemistry, College of Science, University of Jeddah, Jeddah 21959, Saudi Arabia; ijhakeem@uj.edu.sa

**Keywords:** medicinal plants, natural products, phytochemicals and solid cancer

## Abstract

Natural products play a critical role in the discovery and development of numerous drugs for the treatment of various types of cancer. These phytochemicals have demonstrated anti-carcinogenic properties by interfering with the initiation, development, and progression of cancer through altering various mechanisms such as cellular proliferation, differentiation, apoptosis, angiogenesis, and metastasis. Treating multifactorial diseases, such as cancer with agents targeting a single target, might lead to limited success and, in many cases, unsatisfactory outcomes. Various epidemiological studies have shown that the steady consumption of fruits and vegetables is intensely associated with a reduced risk of cancer. Since ancient period, plants, herbs, and other natural products have been used as healing agents. Likewise, most of the medicinal ingredients accessible today are originated from the natural resources. Regardless of achievements, developing bioactive compounds and drugs from natural products has remained challenging, in part because of the problem associated with large-scale sequestration and mechanistic understanding. With significant progress in the landscape of cancer therapy and the rising use of cutting-edge technologies, we may have come to a crossroads to review approaches to identify the potential natural products and investigate their therapeutic efficacy. In the present review, we summarize the recent developments in natural products-based cancer research and its application in generating novel systemic strategies with a focus on underlying molecular mechanisms in solid cancer.

## 1. Introduction

Cancer is a not only a genetic but also a challenging disease and, the main cause of mortality globally. In 2020, there have been a projected 18.1 million cancer cases worldwide. According to the prediction, it is likely to record seven out of ten deaths in Central and South America, Africa, and Asia due to cancer [1,2]. This could be a panic situation that most developing countries prerequisite to upgrade and develop more tactical preparation that limits reconnaissance, early recognition, and active treatment for cancer patients. Several factors such as growing populations, aging, and prompt socioeconomic growth are associated with the rise in cancer burden throughout the world. There is a probability that the cancer maladies rise with increase in age due to the accumulated DNA damage and multi-stage carcinogenesis [3,4,5]. In recent years, due to improvements in novel therapeutics not only diagnosis rate has been increased but also over-all life expectancy of cancer patients has improved [6,7]. 

Presently, the National Cancer Institute (NCI) enlisted 8 categories for treatment of cancer including surgery, radiation, chemotherapy, targeted therapy, immunotherapy, stem cell transplant, and precision medicine [8]. However, the customary approaches for cancer treatment include surgery, radiotherapy, and chemotherapy [9,10]. Conversely, regardless of the various types of chemotherapeutic drugs utilized for the cancer treatment and the remedial triumph of several management programs, the main therapies have not achieved the desired result [11,12]. Some early targeted therapies have shown a positive clinical response, however, frequent drug resistance was observed after an initial positive response in cancer patients. This alteration in the treatment is known as acquired drug resistance, as contrasting to intrinsic resistance, which occurs earlier to any cancer therapy. Acquired drug resistance is developed due to both cytotoxic chemotherapies and targeted therapies with different molecular mechanisms. In most of the cancer, these molecular mechanisms can incorporate compensatory and redundant molecular signaling, targeting mutations developed during treatment, modulation in the expression of the targeted proteins, inhibition of pro-apoptotic pathways, activation of pro-survival signaling, inactivation of DNA repair mechanisms, and upregulation of tumor cell efflux transporters [13,14]. Despite of the progress, resistance to cancer chemotherapy and common side effects are the main issues in patients who have received first-line treatment. Targeted therapy employs a range of small molecules and inhibitors that play an important role in targeting key signaling pathways, result in development of resistance in a rare instance even from first doses. Drug resistance develops in the patient as a result of tumor or cancer cells being certainly chosen for molecular mechanisms that can compensate for the precisely targeted pathway. In light of this, there is an urgent necessity to pursue more selective and active compounds or natural products that have fewer side effects, have more medicinal elements, cost-effective, and have a least level of disease resistance for the management of cancer. However, less evidence exists in the scientific literature about the utilization of such natural products and their mechanism of action against solid cancers. 

Throughout the history, natural products played an important and crucial role in the treatment of human illnesses. Furthermore, traditional remedies, mainly based on native plants, still govern therapeutic practices globally, and natural products cover a huge portion of current-day pharmaceutical tools, especially in the field of antibiotic and cancer therapies. For the management of cancer, timely diagnosis and definitive tumor removal by radiation therapy or surgical resection is the greatest anticipation. Conversely, in case of dealing with malignant and metastatic disease, chemotherapy is usually required. As defined herein, most of the significant improvements that have been recognized for the management of cancer are directly or indirectly associated with the discovery of natural product based chemotherapeutic approach. In past few decades, cumulative evidence has demonstrated the remarkable amplification or utilization of plant-based remedies. As compared with the high cost and side effects of most modern drugs, medical plants have shown significant therapeutic potential with minimal side effects and low cost, such as epigallocatechin gallate (EGCG), resveratrol, curcumin, sugiol etc. EGCG is a polyphenol found in green tea [15,16,17,18].

## 2. Natural Products

Plant-derived natural products are the primary source of biologically active compounds. Moreover, their nontoxic or less toxic nature to normal cells and better toleratation has gained attention from the scientific community and clinicians in the modern drug discovery area [19,20]. The untapped structural diversity of natural compounds is long-lasting importance in drug discovery. It is estimated that the plant kingdom includes at least 250,000 species, of which only 10 percent have been explored for pharmacological applications [21]. Several natural compounds have shown potential activities against metastasis and tumor invasion [16,20,22]. Some plant-derived FDA-approved phytomolecules, such as tetrandrine, lycobetaine, curdione, vincristine, vinblastine, curcumol, monocrotaline, elliptinium, etoposide, gossypol, ipomeanol, taxol, indirubin, 10-hydroxycamptothecin, homoharringtonine, and colchicinamide have shown significant antitumor potential [23]. Approximately more than 600 natural compounds have reported as an anticancer agent. However, keeping in mind the content’s limitation and inability to cover every natural product in a single article, we have briefly discussed some common well-known anticancer compounds such as curcumin, indol-3-carbinol (I3C), resveratrol, kaempferol, epigallocatechin gallate (EGCG), and genistein (Figure 1). 

### 2.1. Curcumin

Curcumin has been suggested as the most potent natural product among the 600 natural products. It is a yellow spice and a phenolic compound derived from the plant Curcuma longa. Curcumin has shown promising chemopreventive and anticancer activity in different cancer models, such as prostate cancer, lung cancer, breast cancer, brain tumors, head and neck squamous cell carcinoma [24,25]. The scientific literatures report the modulation in various signaling pathways by curcumin that results into its antitumor activity [24,26]. The active JAK2 / STAT3 signaling pathway plays a vital role in the initiation and development of various cancers [27]. Thus, JAK2/STAT3 pathway is a well-known therapeutic target for curcumin inhibiting tumor initiation. In primary effusion lymphoma cells, curcumin significantly suppresses the JAK/STAT3 pathway in a dose-dependent manner, which inhibits cell proliferation and induces caspase-dependent apoptosis [28]. Furthermore, curcumin is a more potent inhibitor than a selective inhibitor of AG490 a selective inhibitor of STAT3 phosphorylation of the JAK2/STAT3 signaling pathway in multiple myeloma cells [29]. Scientific studies suggest that curcumin inhibits cell proliferation in numerous cancer cell lines, such as malignant gliomas [30], pancreatic [31,32], hepatocellular [33], ovarian, and endometrial carcinoma [34] by down-regulating the JAK-STAT3 pathway. An in-vivo study reveals that curcumin injected with tumorspheres of lung cancer NCI-H460 cells in nude mice suppressed the tumor growth via repressing the JAK2/STAT3 signaling pathway [35].

### 2.2. Indol-3-Carbinol (I3C)

Another natural product, indol-3-carbinol, mainly present in cruciferous vegetables such as cabbage, cauliflower, and broccoli, also exhibit anticancer activity [36]. It has been reported to inhibit cancer cell proliferation by modulating the expression of insulin receptor substrate-1 (IRS1) and insulin-like growth factor receptor-1 (IGF1R) [37]. A recent study suggested that I3C induces apoptosis in H1299 cells by activating apoptosis signal-regulating kinase 1 (ASK1) [38]. Furthermore, I3C has been reported to exert its anticancer effects through a different mechanism that include decreased cell proliferation, increased apoptosis, and reduced mammosphere formation in MCF-10AR-Her2 cells [39]. Nuclear factor-kappa B (NF-κB) is a master regulator of more than five hundred genes. It plays a crucial role in cancer cell survival by mediating the transcription of several antiapoptotic genes such as *p53*, *p21*, *survivin*, *Bcl-2*, and *Bcl-xL* [40]. I3C and diindolylmethane (DIM) inhibit the activation of NF-κB in SW480 colon cancer cells [41]. Earlier studies also reported that I3C and DIM inhibit the cell cycle in the G1 phase in breast and prostate cancer cell lines [42,43]. Recently it has been shown that I3C decreases cell proliferation and induces apoptosis in the inflammatory breast cancer model. However, this result could not be adequate to evade the development of tumor embolization and metastasis [44]. A study showed that I3C could induce apoptosis in osteosarcoma cells by upregulating the FOXO3 signaling pathway [45].

### 2.3. Resveratrol

Resveratrol is a well-known naturally occurring polyphenol and commonly present in grapes, wine, nuts, berries, and many other human diets [46]. Several studies reported a wide range of pharmacological activities associated with resveratrol such as antiviral, antifungal, anti-inflammatory, antiaging, anticancer, and antioxidant effects [46,47]. Resveratrol has shown anticancer effects in renal carcinoma cells such as ACHN and A498. It reduced cell proliferation, migration, and invasion through inhibition or inactivation of the Akt and ERK1/2 signaling pathways in a concentration-dependent manner [48]. It has also been reported to exert anti-cell proliferation effects through modulation in VEGF expression in an osteosarcoma cell line [49]. In colorectal adenocarcinoma cells (CaCo-2), resveratrol has shown significant growth inhibition at 25 μM due to S/G2 phase arrest through the inhibition of ornithine decarboxylase activity [50]. A study reported that a combination of resveratrol and docetaxel treatment induced apoptosis in prostate cancer cells (C4–2B and DU-145) by inhibiting the cell cycle at the G2/M phase and inducing the expression of pro-apoptotic genes, such as Bax, Bid, and Bak [33].

### 2.4. Kaempferol

Kaempferol, a yellow color compound, is an aglycone type of flavonoid that is made up of glycosides. It contains four hydroxy groups on 3, 5, 7, and 4 positions [51]. The primary sources of kaempferol are fruits, seeds, flowers, leaves, green vegetables, and different plants [52]. It has been reported to be involved in various activities, including anticancer, anti-inflammatory, antioxidant, antitumor, antimicrobial, neuroprotective, and cardioprotective [53]. In addition, kaempferol exerts an anticancer effect in different human cancer cell lines such as SW480, HCT-15, HCT116, HT-29, and LS174-R colon cells [54,55,56]. A study has also shown that treatment of Huh7 cells with kaempferol in hypoxic conditions could inhibit tumor growth through the inactivation of p44/42 MAPK pathways by inhibiting HIF-1α protein [51]. Moreover, kaempferol induced apoptosis in colon cancer cells by activating the upregulation of death receptor 5 and TRAIL receptors [57]. Moreover, a study reported that kaempferol inhibited triclosan and E2-induced breast cancer progression by playing an antagonist role against estrogen receptor and IGF1R signaling [58]. In addition, it also induces apoptosis naturally in MCF-7 cells through the activation of poly ADP-ribose polymerase and via the mitochondrial caspase-9 signaling pathway [58,59]. In vivo study suggests that kaempferol has shown inhibitory activity against metastasis of murine melanoma B16F10 cells and could downregulate the expression of matrix metalloproteinase-9 (MMP-9) and its activity. Therefore it might be a potential anticancer agent for cancer metastasis [60].

### 2.5. Epigallocatechin Gallate (EGCG)

Green tea is a refreshing drink that is used globally. The green tea catechins such as EGCG and other polyphenols showed anticancer activity in different cancer models. EGCG is the most abundant and well-studied catechin found in green tea [40,61,62]. The anti-carcinogenic properties of green tea include controlling cell proliferation, cell death of tumor cells, induction of apoptosis, induction of proapoptotic genes, inhibition of antiapoptotic genes, rise in ROS production and vascular angiogenesis [63]. These catechins modulate the gene expression by directly affecting the transcription factor or indirectly through epigenetic mechanisms [64]. A study revealed that EGCG (10–100 µM) inhibits the receptor activator of nuclear factor-κB ligand (RANKL) and induces NF-kB activity in a murine preosteoclast cell-line RAW 264.7 [65]. Scientific studies have shown significant growth inhibitory potential of EGCG (40–80 µM) in different cancer models, such as prostate, colorectal and liver cancer [66,67,68]. An in vitro study has shown that EGCG and nano-EGCG treatment increases the expression of AMPK phosphorylation in H1299 lung cancer cells [69]. Another study reported the inhibition of cell proliferation and migration in oral cancer cells (H400 and H357) by EGCG treatment through reduced expression of phosphorylated epidermal growth factor receptor (EGFR) [70]. The nano-EGCG regulates various biological activities, including suppressing cell proliferation, inhibiting cell migration, colony formation, and invasion by activating the AMPK signaling pathway in H1299 lung cancer cells [69]. Recently, we have also reported the significant anticancer potential of nano-EGCG in prostate cell lines, viz. 22Rv1 and PC3 [71]. Treating rats with (50 mg/kg) catechin exhibits the downregulation of endotoxin-mediated activation of initial signaling molecule NF-κB, TNFα, nitric oxide, and reactive oxygen species due to catechin’s antioxidant effect [72,73]. In addition, studies have reported that EGCG treatment induced the natural killer (NK) cell activity, triggered the proliferation of B-cell and T cells, and increased NK-cell mediated cytotoxicity in murine leukemia and bladder cancer model [74,75]. 

### 2.6. Genistein

A naturally occurring compound, genistein is an isoflavone belongs to the flavonoid family. It is derived from legumes such as soybeans, lupin and fava beans [17,76]. The consumption of soybeans, lupin, and fava beans is associated with many beneficial effects including lower incidence of some cancers, such as colon cancer, reduction in the cardiovascular disease risk, protection against osteoporosis, and alleviation of postmenopausal symptoms [77,78]. A study reported that genistein inhibits tumor growth and cell proliferation by downregulating the negative effect of epidermal growth factor (EGF) on the activity of forkhead box O3 (FOXO3) in a colon cancer model [79]. In addition, it is also observed that genistein reduces breast cancer stem cells (CSCs) and mammospheres by downregulating the hedgehog-signaling pathway that subsequently regulates cell proliferation, self-renewal ability, stem cell, and progenitor cell maintenance [80,81]. Based on the scientific studies, it is believed that genistein regulates miRNAs expression to stop cell proliferation and up-regulates miR-200 expression, also regulate the essential targets such as vimentin, zinc finger E-box binding homeobox 1 (ZEB1), and slug, which help in the epithelial-mesenchymal transition (EMT) process [82,83]. Genistein could inhibit the tumor development in estrogen receptor alpha (ERα) negative breast cancer through remodeling the chromatin structure in the ERα promoter to reactivate the ERα expression [84]. In the below-mentioned section, we have focused various signaling pathways that are affected by these natural products. 

## 3. Cellular Signaling Pathways as a Therapeutic Target for Cancer Therapy

Cellular signaling are multifaceted communication system consist of three-dimensional molecular cascades containing various signaling proteins. The precious molecular mechanism associated with these proteins are very specific to cell type, cell site, and intra-molecular interactions. Modulation in the homeostasis of these proteins leads to the diverse pathological diseased conditions. However, in case of cancer, alternation in cell signaling and cell communication leads to modulation in the expression of various critical genes associated with the normal functioning of the cells [13]. A series of mutation in numerous cancers associated genes, such as tumor-suppressor and oncogenes lead to cancer proliferation [85]. Early discovery of many oncogenes such as RAS, RAF, MYC, and KIT and several other tumor suppressor genes TP53, PTEN, and BRCA1 have led to the identification of several cancer-associated genetic lesions [86,87]. Currently, cellular signaling pathways and its associated molecular networks are documented for their important roles in regulation of pro-survival cellular processes and are thus predominantly involved in the onset of cancer, and in its prospective management. Many signaling pathways are associated with the development of cancer such as VEGF receptor pathway that activate RAS/RAF/MEK/ERK pathway, and the fibroblast growth factor (FGF) receptor pathway that stimulates multiple pathways, including the PI3K/Akt/mTOR, RAS/RAF/MEK/ERK and act as signal transducer and activator of transcription (STAT) pathways [88,89]. In this review, we have emphasized not only some signaling pathways involved in solid cancer but also highlighted targeted strategies which helps to improve the clinical outcomes. Specifically, two pathways, the PI3K/AKT/mTOR signaling pathway and Ras/MAPK pathway, are repeatedly activated, or mutated in many solid cancers. 

PI3K/Akt/mTOR pathway plays an important role in the regulation of several normal cellular activities that are also important for tumorigenesis such as cell survival, migration, cell cycle progression, angiogenesis, and EMT [16,90]. Aberrant regulation or activation of the PI3K/Akt/mTOR cascade is mainly involved in the development of various human cancers such as acute lymphoma (AML), T cell acute lymphoblastic leukemia (T-ALL), breast cancer, ovarian cancer, prostate cancer, and mantle cell lymphoma [91,92,93,94]. The activation of the PI3K/Akt/mTOR pathway begins in the response of extracellular stimuli and growth factors leading to the activation of receptor tyrosine kinases (RTKs) that make the autophosphorylation of tyrosine residues and transphosphorylation of adaptor proteins [94]. The phosphatidylinositol (PI)-3-kinase (PI3K) class Ia activation occurs as its Src homology (SH2) domains bind with the p85 regulatory unit to specific phosphotyrosine residues on the activated receptor or associated adaptor proteins, which helps the enzyme to move from cytosol to the plasma membrane and activates the p110 catalytic unit [16,95]. Activated PI3Ks acts as lipid kinase and phosphorylate phosphatidylinositol 4,5-bisphosphate (PIP2) to produce the phosphatidylinositol-3,4,5-triphosphate (PIP3) which further plays a crucial role in the form of second cellular messenger to control the cell growth, proliferation, and cell survival [16,96]. PIP3 starts the Akt activation through the recruitment of PDK-1 and the PKB on the plasma membrane, where PDK-1 makes the phosphorylation at Threonine(T)308 residue of Akt in the activation loop [97,98]. Consequently, Akt gets activated and moves to the cytosol and nucleus where it phosphorylates the different substrate downstream proteins including Bcl-2 associated agonist of cell death (BAD), forkhead box class O (FoxO) and glycogen synthase kinase-3 (GSK3) α/β to support the cell growth, survival, and other cellular effects [99]. Furthermore, Akt indirectly activates its downstream target mTOR by phosphorylating and inhibiting tuberous sclerosis complex 1 and 2 (TSC1/2) at S939 and T1462 residues (Figure 2). Thus, mTOR positively regulates different cellular functions by promoting protein synthesis and inhibition of autophagy by releasing its inhibitory effects on Ras-related GTPase Rheb complex [100,101].

Another signaling pathway, the mitogen-activated protein kinase (MAPK) is comprised of different signaling cascade components and has been observed to be deregulated in various human cancer. The hyperactivation of Ras/RAF/MEK/ERK (MAPK) pathways is noted in more than 40% of human cancer cases [102]. A series of activated kinases send the extracellular signals to regulate the various cellular activities, including cell proliferation, differentiation, apoptosis, cell growth, and cell migration. It is reported that abnormal or aberrant activation of RTKs or gain-of-function mutations in the RAS or RAF genes are the leading causes of alteration in RAS-MAPK in human cancer. For these reasons, the RAS-MAPK pathway is a well-established therapeutic target for cancer treatment and its management [103,104]. In resting cells, plasma membrane-associated Ras-GDP remains inactive with RAF, MEK, and ERK in the cytosol. However, in response to the exposure to extracellular stimuli (growth factors, hormones, and cytokines), RTK autophosphorylation begins to generate the binding sites for SHC and GRB2 adaptor molecules that recruit SOS and RasGEF (GTPase exchange factor) to the plasma membrane (Figure 3) to trigger the activation of RAF/MEK/ERK kinase cascade [105]. Furthermore, activated ERKs stay in the cytoplasm or move into the nucleus where they regulate the various physiological processes by phosphorylation of several substates [105,106,107].

These pathways are prominently interconnected in facilitating upstream signals from receptor tyrosine kinases (RTKs) to intracellular effector proteins and cell cycle regulators [108]. Interestingly, the signals transmitted from the extracellular space into the cytoplasmic and nuclear compartments, the PI3K and MAPK pathways are intensely connected via several positive and negative axis. Additionally, other signaling pathways linked with the process of EGFR activation are phospholipase C-g and the JNK. These molecular pathways participate primarily in processes of cell proliferation, cell migration, and transformation. Together, these signaling pathways regulates gene transcription, cell proliferation, cell cycle progression, survival, adhesion, angiogenesis, and cell migration in solid cancer [102,109]. Various kinases have been observed to be meticulously participate in the processes of tumor cell proliferation and survival [110]. Modulation in the RTK activity is the key mechanism of the tumor cells to escape from physiological controls on survival and growth. Atypical activation of RTK takes place due to receptor over-expression, gene amplification, mutations, and abnormal receptor regulation associated with the development of various forms of cancer in human [111]. In many solid cancers, the family of RTKs has been observed to be deregulated, leading to not only overexpression and amplification of EGFR but also unsuitable cellular stimulation. Receptor overexpression has been associated with a more aggressive clinical outcomes in numerous solid tumor types [112]. Most of the drug resistant and many chemotherapy-naive tumors are characterized by deregulated RTK signaling. Furthermore, pan-cancer analyses have exhibited rearrangements in chromosome due genomic instability, as the initial events in many cancer types like melanoma, glioblastoma, breast, and adenocarcinoma [86]. Moreover, the survival of cancers expressing hormonal receptors, such as prostate, breast, and ovarian cancers essentially depend on the growth signal induced by their relative hormones, such as estrogen and androgen. For the treatment per se of solid cancer a variety of anticancer drugs have been developed. These chemo drugs include the cytotoxic, cytostatic agents, and newer compound that interfere or impede with intracellular processes of solid cancer [113]. Some essential cancer-causing pathways and targets of natural products are presented in Figure 4.

## 4. Scientific Principles Related with Cancer Chemoprevention

Chemoprevention is the application of pharmacological or natural compounds for the inhibition of invasive cancer. It integrates the concept of delay which infers several years, or decades that might be added to human life cycle. Scientific interest in cancer biology research, especially in the area of chemoprevention has significantly improved with the advancement in the understanding of carcinogenesis and identification of potential molecular targets associated with this process. The process of carcinogenesis has been recognized as a clonal propagation and accumulation of genetic damage over the period. Chemopreventive compounds are potent to interrupt clonal propagation in abnormal cells by delaying DNA damage, impeding, or reversing the malignant phenotype, or promoting apoptosis in the impaired cells of premalignant lesions [114]. The importance of chemoprevention further improved by achieving the control of breast, colon, and prostate cancer. There are more than 10 natural product-based medications are approved by USFDA for the decline in cancer risk [115]. Recently, the concept of chemoprevention was considered as a vital and enthusiastic strategy for controlling solid cancer. It does not only play an essential role in preventing the growth of the invasive and metastatic potential of cancer (neoplasm) but also lowers the cancer prevalence rate. Chemoprevention strategy can be categorized into three parts: (1) primary prevention, impeding the growth of tumors in healthy individuals; (2) secondary prevention, inhibiting the growth of tumors in those individuals with precancerous lesions like invasion; (3) tertiary prevention, inhibiting recurrence or relapse of cancers in target patients [116]. Besides rigorous biomolecular validation, chemopreventive compounds must retain very less or no toxicity because they will be applied by an essentially healthy people at high cancer riskof. Numerous classes of compounds like cyclooxygenase (COX) inhibitors, retinoids, and sex hormone antagonists are very useful in the prevention of various epithelial cancers [117,118,119]. Similarly, various molecular mechanisms of action have been defined and efforts have been made to broadly classify the chemopreventive compounds according to different stages of carcinogenesis [120]. Nonetheless, it is likely that many compound, mainly derived from dietary ingredient have multi-targeted effects throughout the carcinogenic process. Agents that prevent cancer initiation are customarily termed ‘blocking agents’. They work by reducing the interaction between chemical carcinogens and DNA, thus reducing the level of damage [121]. Once beginning has arisen, chemopreventive compounds might impact on the promotion and progression of initiated cells; such compounds are frequently termed ‘suppressing agents’ [110]. Recent studies demonstrated that intervention in tumor metabolism and energy homoeostasis through AMPK and mTOR signaling pathways may be a striking tool for chemopreventive agents [122]. 

## 5. Role of Natural Products in the Management of Cancer

Since ancient times, local communities consume natural products and herbs in health care system for preventing several diseases, including cancer [20]. Approximately 40% of alternative therapies including natural product based herbal medicine were used for preventing various disease in the United States of America (USA). Generally natural products, as part of complementary medicine in the USA, have provided a basis to conduct the research for discovering novel plant-based medicinal agents, and more than half of currently existing drugs are based on natural products [123]. The epidemiological data also suggested that more than 50% of the approved anticancer agents are either natural compounds or natural product derivatives derived from herbal medicine [124,125]. 

Despite substantial development in the prevention and treatment of cancer, major gaps still exist, and further progresses are still required. Various studies have suggested that substantial application of plant-based therapeutics with ability to regulate physiological functions including flavonoids, phenolics, alkaloids, and organosulfur compounds, have been recognized to inhibit cancer in several in vivo and in vitro cancer models via different mechanisms [126,127,128,129,130,131,132,133]. However, very few evidence-based studies exist in the scientific literature regarding the application of biological natural agents and their associated molecular mechanism against solid cancer. Using modern technology and novel research strategies, more plant-derived components have been discovered for the management of advanced-stage cancer without significant collateral damage. 

Natural product acquired from diverse sources indicates the ability to modulate numerous physiological signaling pathways such as apoptosis, metastasis, angiogenesis, and drug resistance (Figure 5), necessary for the treatment of cancer [127,134,135,136].

Therefore, it is imperative to apply various strategies for the management of this deadly diseases by using natural products, especially phytochemicals [137,138]. Various studies suggested the anticancer activity of natural products-based extract and its active ingredients in in-vitro cancer cell line models and pre-clinical animal models of many solid cancers [139,140,141,142,143,144,145,146,147,148,149,150,151]. Nature remains to be a rich source of biologically active and diverse chemotypes. Unfortunately, a small percentage of natural products are being developed into clinically (Table 1) effective drugs via exploitation of chemical techniques such as metabolomics, alteration of their biosynthetic pathways, and total or combinatorial fabrication [16,152].

Moreover, recent developments in the formulation strategies of novel biological active compounds may results in the more efficient application of drugs to the cancer patients. There are several tools such as fusion of toxic natural molecules to monoclonal antibodies and polymeric carriers precisely targeting epitopes on the membrane of targeted tumor cell result in the discovery and development of more active antitumor drugs [165]. Additionally, the scientific inputs and multidisciplinary alliances among various researchers around the globe are also required to optimize the most active biological compounds at the molecular level ultimately leading to significant control of cancer progression [147,148,149].

## 6. Conclusions and Future Perspectives

The high rate of mortality and morbidity in solid cancers remain a primary task for scientific investigation. Still conventional tools, such as surgery, radiotherapy, and chemotherapy are effective in some patients but five-year survival rate in solid cancer patients is usually miserable. In most of the cases chemotherapy causes drug resistance and unwanted toxic side effects in the patients. In last decade, plant-derived natural compounds have been used as a potent candidate for the management of cancer. This review provides a deep understanding about the role of plant derived natural products in the treatment of cancer by modulating various signaling pathways. Natural compounds are evolving as a prospective therapeutic tool in cancer biology research owing to their easy accessibility and cost-effectiveness. Various natural agents are used in preclinical or clinical settings for the management of cancer [75,76]. Several epidemiological data suggest that high nutritional intake of fruits and vegetables reduce the risk of cancer [77]. The scientific evidence described in this article highlight the uninterrupted development and advancement in the field of plants based natural products research, demonstrating that it occupies a critical position in the use of chemopreventive compounds. The discovery and development of anticancer agents have steadily shifted from all those drugs having a single target and robust side effects to natural plant-based drugs with less or no toxicity. Most of the natural medicine typically affects more than one pathway (for instance triggering apoptosis, inhibiting cell proliferation, etc.). The multitarget potential of natural products allows them to efficiently offset the biological complication in cancer and offer favorable resources for cancer chemoprevention. Better understanding of cellular signaling pathways and its regulation may demonstrate a valuable strategy in cancer therapy. To discover the effective treatment approaches with least side effects and low cost, the researchers are encouraged to conduct research on natural resources, exclusively on plants and their active constituents. Therefore, it is essential to enhance the mechanistic and clinical based studies to discover novel and effective natural chemopreventive compounds. We believe that natural antitumor active ingredients and precursor drugs could provide an alternative or adjuvant treatment strategies in clinical medicine for the cure of solid cancer.

## Figures and Tables

**Figure 1 cells-11-02209-f001:**
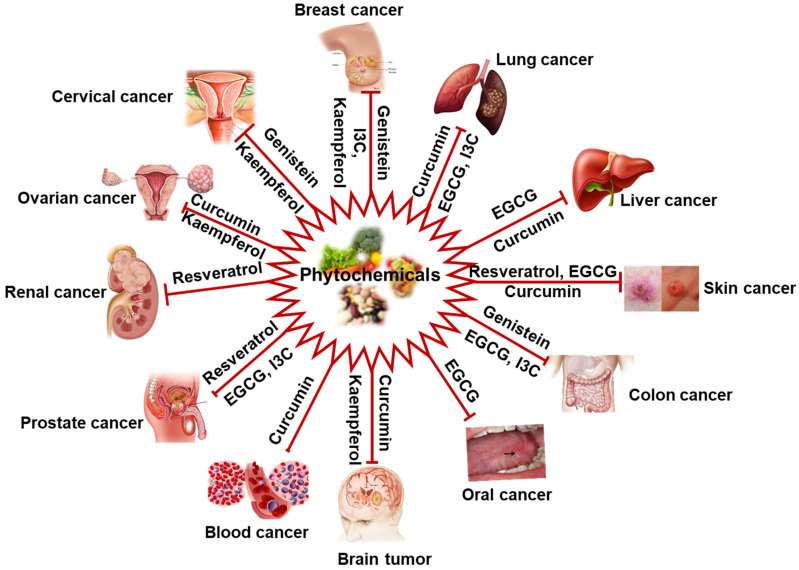
Schematic representation of cancer types that could be prevented/managed by natural products (phytochemicals). IC3, Indol-3-carbinol.

**Figure 2 cells-11-02209-f002:**
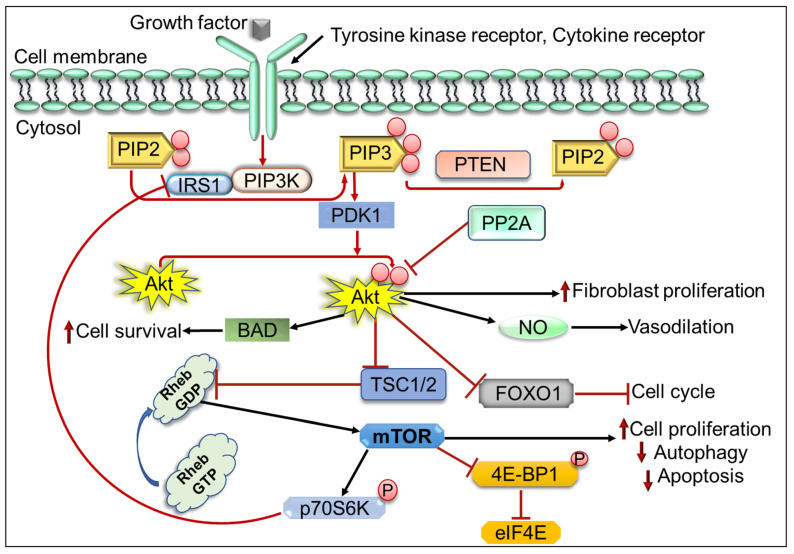
Schematic overview of PI3K/Akt/mTOR pathway. BAD, BCL2 associated agonist of cell death; FOXO1, Forkhead box O1 protein; IRS1, Insulin receptor substrate 1; 4EBP1, Eukaryotic translation initiation factor 4E-binding protein 1; p70S6K1, p70 Ribosomal S6 kinase 1; PIP2, Phosphatidylinositol 4,5-bisphosphate; PTEN, Phosphatase, and tensin homolog deleted on chromosome 10; PDK1, 3-Phosphoinositide-dependent kinase 1; PP2A, Protein phosphatase 2A; Rheb GDP, Ras homolog enriched in brain GDP; Rheb GTP, Ras homolog enriched in brain GTP and TSC, Tuberous sclerosis complex.

**Figure 3 cells-11-02209-f003:**
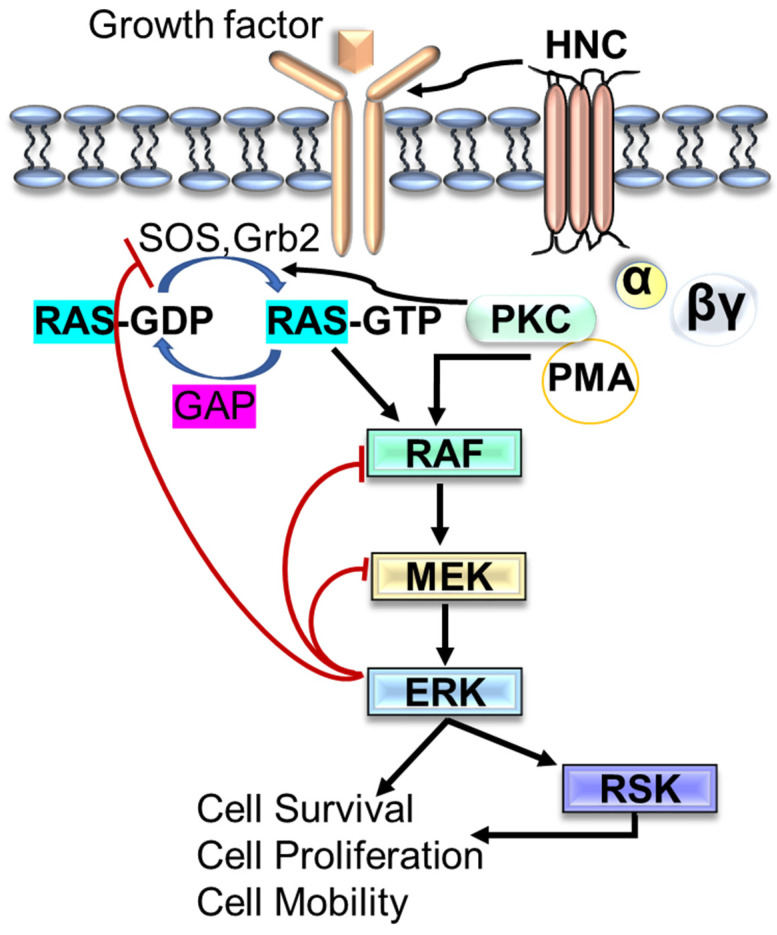
Schematic representation of Ras/MAPK pathway. ERK, Extracellular signal-regulated kinase; GAP, GTPase-activating protein; PKC; Protein kinase C; PM, Phorbol 12-myristate 13-acetate; RSK, Ribosomal s6 kinase.

**Figure 4 cells-11-02209-f004:**
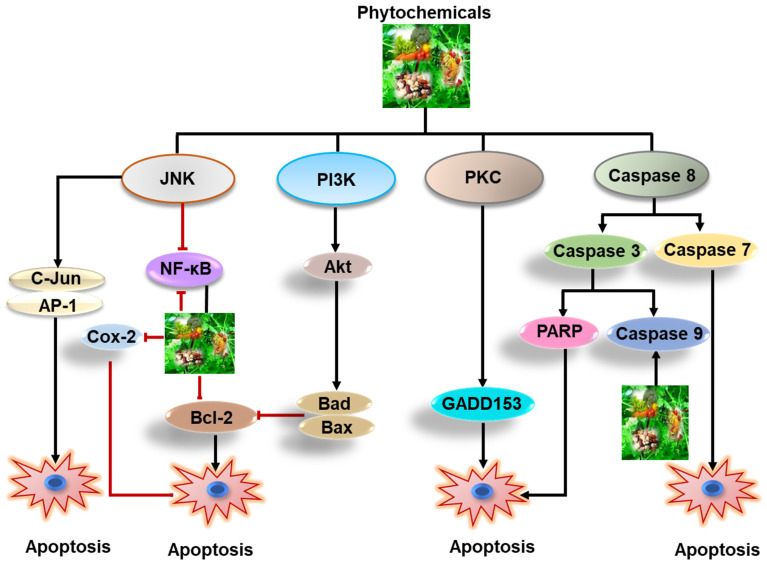
Depiction of important cellular pathways regulated by natural products that could be utilized for therapeutic purpose in solid cancer.

**Figure 5 cells-11-02209-f005:**
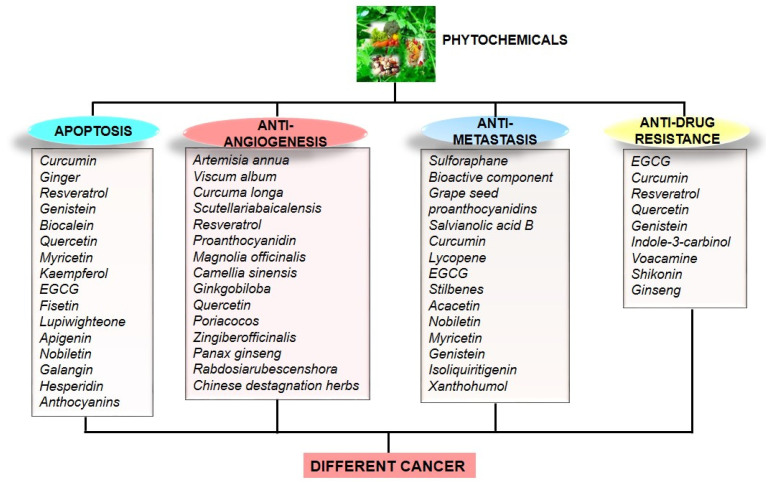
Schematic diagram of cellular process regulated by different phytochemicals against various cancer forms.

**Table 1 cells-11-02209-t001:** A list of phytochemicals as anticancer agents for different cancer in clinical trial.

Phytochemicals	Clinical Trial Type and Phase	Cancer/Conditions Type	References
Quercetin	For prevention, Phase not applicable	Prostate cancer	[84]
	For prevention, Phase II	Squamous cell carcinoma	[84]
Green tea catechins	For prevention, Phase II	High breast density and postmenopausal in women	[150]
	For treatment, Phase II	Neoplasm and multiple myeloma	[150]
	For treatment, Phase II	Oral premalignant lesion	[153]
	For treatment, Phase II	Bladder cancer	[150]
	For prevention, Phase II	Tobacco use disorder	[150]
	For treatment, Phase I	lung carcinoma	[150]
Green tea polyphenon E and Erlotinib	For prevention, Phase I	Lesions of head and neck cancer	[154]
Curcumin	For prevention, Phase II	Familial adenomatous polyposis	[155]
	For treatment, Phase I	Advanced osteosarcoma	[150,156]
	For treatment, Phase II	Advanced pancreatic cancer	[157]
	For prevention, Phase I	Colon cancer	[150,158]
Indole-3-carbinol/3,3-diindolylmethane (IC3/DIM)	For prevention, Phase II	Prostate cancer progression	[159]
	For treatment, Phase II	Breast cancer	[150]
	For prevention, Phase I	Women carrying BRCA1 mutation	[160]
(IC3/DIM) + Radical prostatectomy	For treatment, Phase I	Prostate cancer	[150,161]
Genistein	For prevention, Phase II	Patients with bladder cancer	[162]
Resveratrol	For prevention, Phase II	Colorectal cancer	[163]
Betulinic acid	For treatment, Phase I/II	Dysplastic nevi that can be change into melanoma	[164]
Ingenol mebutate	For prevention, phase I/II	Human non-melanoma skin cancer	[164]

## Data Availability

Not applicable.

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
