# Peer review of "The Role of Natural Products and Their Multitargeted Approach to Treat Solid Cancer"

_cells, 2022, doi:10.3390/cells11142209_

Round 1

Reviewer 1 Report

Although the authors improved the overall quality of the manuscritpt, several concerns should be considered. 

They should deepen their dissertation, focusing better on the evidence available on the natural compounds analyzed.

A table including the clinical trials currently ongoing could be useful.

Authors could also focus on certain cancer types. 

As previously indicated, I suggest, but not lilimted to, the following manuscripts: doi: 10.3390/nu13113750; doi: 10.3390/nu12092648.

Reviewer 2 Report

Congratulations 

Reviewer 3 Report

Examples of anti-cancer phytochemicals have been included in the text. However, the content on phytochemicals themselves, which should be the focus of the review is very brief and lacks details. For example, the authors only spend 4 sentences on curcumin, for which a rich literature exists. The manuscript is still more a review of oncology rather than about phytochemicals. The mechanisms of example phytochemicals' effect on cancer should be expanded to specify the molecular pathway involved for each chemical in each cancer type. Figures are generally broad and lack specifics. Figure 5 is not cited. 

Round 2

Reviewer 1 Report

Authors addressed exaustively my comments

Reviewer 3 Report

The article has been greatly improved.

This manuscript is a resubmission of an earlier submission. The following is a list of the peer review reports and author responses from that submission.

Round 1

Reviewer 1 Report

It’s very important to find new drugs and combination therapies to fight solid cancer and this review gives an important suggestion about what to use to improve the modern strategies. The use of natural products as adjuvants or sometimes as a real therapy is an ancient but an established strategy to prevent or treat solid cancers. Furthermore, a large part of the conventional chemoterapeutics have a natural origin and it is demonstrated by a number of clinical trials (RCT) that natural products have a multitarget action against the pathways causing or promoting solid cancers. The limit of this review is the lack of concrete examples proving the efficacy of natural products in each type of solid cancer mentioned.

My suggestions:

The use of the word “vital” in the title could be avoided

Revise keywords adding for example phytochemicals or nutraceuticals. The keyword “Molecular mechanism of action” is misleading because the review only talks superficially about the mechanism of action of natural products and no medical plants are mentioned in detail

Simplify lines 31-33.

Lines 53-60 could be removed. There is some elementary information.

The manuscript is a general dissertation that should be extensively revised. Authors could add the mechanism of action of some compounds, add results of clinical trials and etc. Focus on some cancer types. I kindly suggest the following manuscript https://doi.org/10.3390/nu13113750

No concrete examples to support the fact that “as compared with synthetic agents, medical plants have therapeutic application with less side effects and inferior cost”

Reading the review it’s not clear how the mentioned signalling pathways are inhibited/activated by natural products

In the section “Role of natural products in the management of cancer” is said that various studies suggest the anticancer activity of natural products … in cell line models and pre-clinical animal models of many solid cancer. This evidence should be supported by RCTs.

The article says that five- years survival rate in solid cancer patients is miserable but this is not a generalizable conception, for example this is not true in prostate cancer

The article makes no difference between potential new natural drugs to be used as a single therapy and natural adjuvants to be associated with conventional therapies

 Considering that the present manuscript is not a systematic review, the paragraph on review methodology is not really pertinent.

Figures could be improved adding novelty and originality, focusing on novel compounds and etc.

The scientific soundness should be highly increased.

Reviewer 2 Report

I feel the authors choose a broad subject 'The Vital Role of Natural Products and its Multitarget Based Treatment of Solid Cancer' for the review and did not do justice to it. 

1) Not a single anti-cancer molecule like curcumin, withaferin A, mahanine, celastrol, etc derived from natural sources was discussed.

2) Generalized headings like the Role of natural products in the management of cancer do not reveal much information to the readers about the topic.

3) Sessions like Review methodology is not required since everyone knows how to gather the information from search engines. It just fills the space in the manuscript.

4) Information provided under 'Cellular signaling pathways as a therapeutic target for cancer therapy is superficial and does not reveal much about the molecular and cellular signaling pathways. 

5) Figures need a lot of improvement. I would suggest using software like Biorender to depict the pathways.

6) Overall the manuscript needs to be revised drastically. 

Reviewer 3 Report

This manuscript by Muhammad et al aims to “summarize the recent development in natural products based cancer research and its application in generating novel systemic strategies with a focus on underlying molecular mechanisms”. Unfortunately, it fails to deliver the information on both aspects. The article reads more like a popular science assay that introduces cancer biology and the concept of natural products to the general public. The authors spend most of the content on cancer biology, with only section #5 talking about natural products. Even in section #5, the authors only give a superficial discussion of the general mechanism by which natural products can affect cancer. The discussion lacks depth and specifics of cancer-related natural products. The manuscript is a big deviation from the title and the purported aim of this review.